# Elucidating the causal mechanisms of *Taenia solium* transmission in humans, pigs and the environment: A global systematic review

Elias Jackson[1,2,3], Veronique Dermauw[4], Guy Eyaba Tchamdja[1],
Mohammad Shah Jalal[1,2,3], Katrina Di Bacco[1,2,3], Pierre Dorny[4], Amanda Janitz[5],
Hélène Carabin[1,2,3,6]*

**1** Department of Pathobiology and Microbiology, Faculté de Médecine Vétérinaire, Université de Montréal, Saint-Hyacinthe, Canada, **2** Centre de Recherche en Santé Publique, Montréal, Canada, **3** Groupe de Recherche en Épidémiologie des Zoonoses de Santé Publique, Université de Montréal, Saint-Hyacinthe, Canada, **4** Department of Biomedical Sciences, Institute of Tropical Medicine, Antwerp, Belgium, **5** Department of Biostatistics and Epidemiology, University of Oklahoma Health Sciences Center, Oklahoma City, Oklahoma, United States of America, **6** École de Santé Publique, Université de Montréal, Montréal, Canada

* helene.carabin@umontreal.ca

## Abstract

### Background

*Taenia solium* cysticercosis/taeniasis disease complex is a multifaceted neglected tropical zoonosis. Previous reviews on *T. solium* cysticercosis/taeniasis risk factors have been limited by geographic region and/or host, making it difficult to understand the complex web of causes underlying infection at the various stages of its life cycle. Our objective was to elucidate the causal mechanisms involved in the transmission of *T. solium* to all hosts through a systematic review of the literature.

### Methodology/Principal findings

We conducted a systematic literature search about the epidemiology of *T. solium* infection published before May 2020 (OpenScience protocol https://doi.org/10.17605/OSF.IO/U64K3). We searched PubMed, Web of Science, and CABAbstracts, and reference lists of systematic reviews identified in our searches for relevant studies meeting the inclusion criteria: i) describing a risk factor-outcome association related to *T. solium* infection, and ii) published in English, French, Spanish, or Portuguese. Qualitative or quantitative epidemiologic associations were extracted and evaluated according to a modified version of the Quality Assessment Tool for Quantitative Studies. Our review is reported according the PRISMA guidelines. We extracted 876 associations from 159 studies. Risk factors of human cysticercosis are well-studied, while taeniasis is least-studied (10% of associations). The evidence suggests that male gender is a risk factor for human cysticercosis while female sex is a risk factor

**Data availability statement:** The authors have provided all data extracted from published articles in our supplemental files. No original data from research subjects was used as part of this systematic review.

**Funding:** This work was supported by L'Institut de Valorisation des Données (E.J., grant number: PhD-2019a-2683768193) and by the Canada Research Chair in Epidemiology and One Health (H.C., grant number: CRC 950-231857). The funders had no role in study design, data collection and analysis, decision to publish, or preparation of the manuscript.

**Competing interests:** The authors have declared that no competing interests exist.

for porcine cysticercosis. The direction of association did not differ according to the quality of the studies across most studied risk factors.

## Conclusions/Significance

Our rigorous systematic review is the first to consider risk factors of human and porcine *T. solium* disease simultaneously, allowing a synthesis of information from both hosts in order to elucidate causal mechanisms for risk factors of cysticercosis. Our work also highlights areas of *T. solium* epidemiology which are understudied, such as the causal mechanisms of taeniasis.

### Author summary

*Taenia solium* is a parasite with a complicated life cycle involving humans and pigs. We reviewed the literature using strict criteria to assess what is known about the risk factors for infection in each stage in the life cycle of the parasite. By considering the risk factors for humans and pigs simultaneously, we improve our ability to identify which risk factors are involved in acquiring the different stages of the infection beyond what we could learn by studying them individually.

## Introduction

*Taenia solium* cysticercosis/taeniasis (TSCT) is a parasitic, zoonotic infection complex, mainly endemic in regions where pigs are allowed to roam freely and sanitation is poor. Pigs act as intermediate hosts to *T. solium* by ingesting eggs shed by a human tapeworm carrier. These eggs develop into larval cysticerci that live within different tissues (particularly skeletal muscle and the brain), causing the disease cysticercosis. Humans become definitive hosts to the adult tapeworm (taeniasis) when they eat undercooked pork containing viable cysticerci. The adult tapeworm establishes in the small intestine and produces eggs that are shed into the environment, completing the life cycle. Humans can also act as accidental hosts by ingesting the eggs, upon their development in larval cysticerci (human cysticercosis) [1] (Fig 1).

While neither porcine cysticercosis nor human taeniasis are associated with significant symptoms, human cysticercosis can cause chronic neurologic symptoms (most commonly epileptic seizures and severe chronic headaches) when the cysts develop, degenerate and die in the central nervous system [2]. Although some parasiticides can kill the adult and larval stages of *T. solium* in any of its hosts, the symptoms caused by human cysticercosis may persist even after the larvae have degenerated, which may require lifelong management [3]. Unfortunately, areas where *T. solium* is endemic are often the same areas where access to diagnostic and medical services is limited. Thus, it is essential to prevent infection in the various hosts to limit the burden caused by cysticercosis.

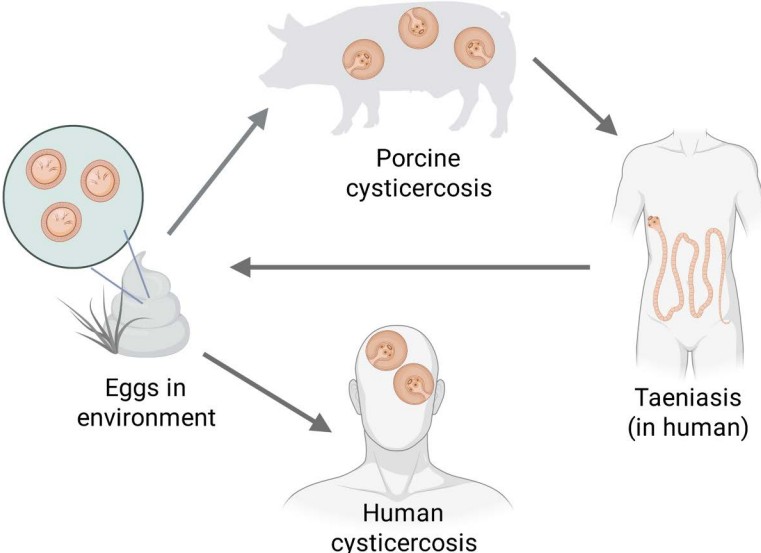

**Fig 1. The life cycle of *Taenia solium*.** Created in BioRender. Jackson, E. (2026) https://BioRender.com/mqrbym7.

TSCT has been deemed a potentially eradicable disease [4] in spite of the complicated multi-host life cycle of the parasite causing it, because the involved hosts are either human or domesticated by humans and thus easily reached by interventions [5]. Many possible eradication methods have been proposed. However, since resources to intervene are limited, interventions must be prioritized by their efficacy, sustainability and low cost.

To understand the epidemiology of TSCT, one must understand not only the immediate causes of TSCT transmission (e.g., free-roaming pigs for porcine cysticercosis or not cooking pork thoroughly for taeniasis), but also what factors are leading to these causes. Currently, there are no global systematic reviews of TSCT that simultaneously consider factors related to human cysticercosis, human taeniasis, and porcine cysticercosis. Recent reviews summarized prevalence and geographic distribution of disease in one species [6,7], surveyed control programs [5], or only considered causation in one species and one region [8]. This fragmented view makes it difficult to understand the multifactorial causes of *T. solium* transmission.

The primary objective of this systematic review is to identify the risk factors of *T. solium* infection in humans, pigs, and the environment simultaneously in order to summarize what has been studied about each risk factor and highlight understudied risk factors. By considering the risk factors of human and porcine cysticercosis simultaneously, we aim to elucidate some of the causal mechanisms of each form of this disease by extrapolating what is known about one form of infection and applying it to the other.

## Methods

A systematic literature search was performed to identify studies of TSCT epidemiology published on or before May 2020. This literature search is published under OpenScience protocol (https://doi.org/10.17605/OSF.IO/U64K3). The systematic review included two levels. At the first level, studies found in the literature were reviewed using a set of eligibility criteria. At the second level, associations between each possible risk factor and an outcome linked to T. solium infection were reviewed with additional set of eligibility criteria.

At the study level, relevant articles were identified via two methods. Firstly, the bibliographic databases PubMed, Web of Science, and CABAbstracts were searched using the terms ("Taenia solium" AND "epidemiology"). No filters were applied to limit the year of publication, type of study, language, or any other variable. Secondly, the reference lists of all systematic reviews identified during the bibliographic database search were screened to identify articles with risk factor-outcome associations related to TSCT not identified in the initial search. Systematic reviews retrieved via the bibliographic databases search were identified as such based on the mentioning of "systematic review" in their title or abstract or through the description of the methodology. Narrative reviews were not searched for references. All articles included in the reference lists of these systematic reviews underwent the same systematic review process as that for articles identified through the bibliographic databases. Duplicate articles identified from the database search or from the reference lists of systematic reviews were removed. The remaining titles and abstracts and full texts of all articles identified by the two methods were reviewed using the exclusion criteria shown in Table 1. Titles and abstracts that were in English or French were evaluated by teams of two reviewers (combination of EJ, VD, AJ, HC, and MSJ) to determine their eligibility. Reviewers were not blinded to the decisions of other reviewers at this stage. Studies in Spanish and Portuguese were reviewed by a single reviewer (HC). If the two reviewers disagreed as to whether a study was eligible based on the title/abstract, then the study was reviewed in full to ensure that no information was excluded prematurely. The full text articles were reviewed in the same way as the abstracts except for disagreements which were settled by a discussion between the two dissenting reviewers and, when an agreement could not be reached, through discussion with a third mediating reviewer.

**Table 1. Detailed description of exclusion criteria for studies used in the systematic review of the identified articles.**

| Exclusion criteria | Explanation |
| --- | --- |
| Unable to find | Unable to locate the full text of the article despite requesting from multiple institutions |
| Duplicate | Article is a duplicate of data already identified in the literature search. This includes a few studies which were published in greater detail in a later publication. |
| Wrong article type | Article is not an original research article discussing or measuring the association between two variables. Examples of excluded articles include review articles and case reports. |
| Wrong outcome | The article did not use cysticercosis and/or taeniasis as caused by *T. solium* as the outcome of interest. Examples of excluded articles include those that examined the prevalence of *T. saginata* and those that considered only cases of neurocysticercosis. |
| Prevalence-only | The article reported the prevalence of TSCT in a population but neither tested any potential associations nor provided information in population subgroups to evaluate potential associations. |
| Symptomatic or deceased populations | The sample was chosen from a population with symptoms of disease. Examples of excluded articles include those that sample from patients hospitalized for seizures or those that measure prevalence at autopsy. |
| Wrong language | The study was not in a language readable by anyone in our group (i.e., not in English, French, Spanish, Romanian, or Portuguese). |

For each study included in full-text review, information about the country where it took place, whether the setting was rural or urban, and on the design (case-control, cross-sectional, etc.) was extracted. The World Income Level classification was used to determine the relative poverty of the countries where the included study were conducted [9]. Countries were also grouped according to their World Health Organization regions [10]. Data on country region was summarized in a waffle plot created in R [11] using the now-archived package waffle (https://cran.r-project.org/web/packages/waffle/index.html)

At the second level of the review, for each included study, all associations between any two variables were evaluated. These association may be between a risk factor and an infection status (i.e., human cysticercosis, human taeniasis, porcine cysticercosis, taeniid eggs in the environment) or not consider *T.* solium at all (e.g., an association between education level and current pig raising [S3 File]). Eligible associations included quantified measures of association, statements from the authors of statistical significance or its lack without accompanying quantified measures of association, qualitative results from focus groups, or even author observations in the discussion. Associations between a known transmission method and disease state were excluded unless the study evaluated some modification of that risk factor (e.g., pork eating as a risk factor for human taeniasis was excluded, but eating pork at the market was included). We also excluded associations involving symptomatic presentations of TSCT (e.g., associations between pork eating and seizures in humans), in harmony with our exclusion of symptomatic populations at the study level (Table 1). Associations presented in such a way that we could not determine their direction or magnitude were excluded. Multiple associations could be evaluated for inclusion from one study. A single study could also have multiple included associations for the same proposed cause-effect relationship if that study defined the variables in different ways (e.g., assessing latrine availability by whether the house had a latrine and by whether the participant reported being able to use a neighbor's latrine).

All associations included after the second level of review were graded for their quality of evidence using a modified version of the Quality Assessment Tool for Quantitative Studies [12]. At this third level, the quality of evidence was graded as strong (likely free from significant bias), moderate (significant bias possible in one aspect of the study), or weak (significant bias possible in multiple aspects of the study). This tool, originally meant to evaluate studies, was adapted to assess the quality of each association. Hence, strong, moderate and weak quality associations may occur in the same publication. For example, a study may have assessed the association between risk factors and both taeniasis and human cysticercosis, but the test for taeniasis may have been of poor validity while the test for human cysticercosis had high validity and reliability. A detailed description of our modified tool can be found in S1 and S2 Files. The form itself, as well as the ratings for each extracted association, are found in S3 File. Each association was evaluated by two authors blinded to each others' ratings (EJ, AJ, GET, MSJ, KD, and HC). Bias related to the exposure or outcome was assessed for all associations by EJ.

After these levels of review, our included associations were visualized according to the cause-effect outcomes studied or discussed. We visualized, the number of included associations, and our quality assessment for included associations using a lollipop chart generated in R [11] using the package tidyverse [13]. We summarized in barcharts from Excel the trends and direction of evidence for each group of exposure-outcome association with at least five extracted associations to determine the plausibility of the association being causal and whether the exposure seemed to be protective or a risk factor. We did not summarize our results quantitatively because some of our extracted information is non-quantitative (study author comments on significance with no data presented, for example.)

## Results

### Review process and description of included articles

Our search identified 2,372 studies, of which 470 underwent full-text review. Of these, 159 studies met our inclusion criteria. Fig 2 shows the results of our full-text selection process.

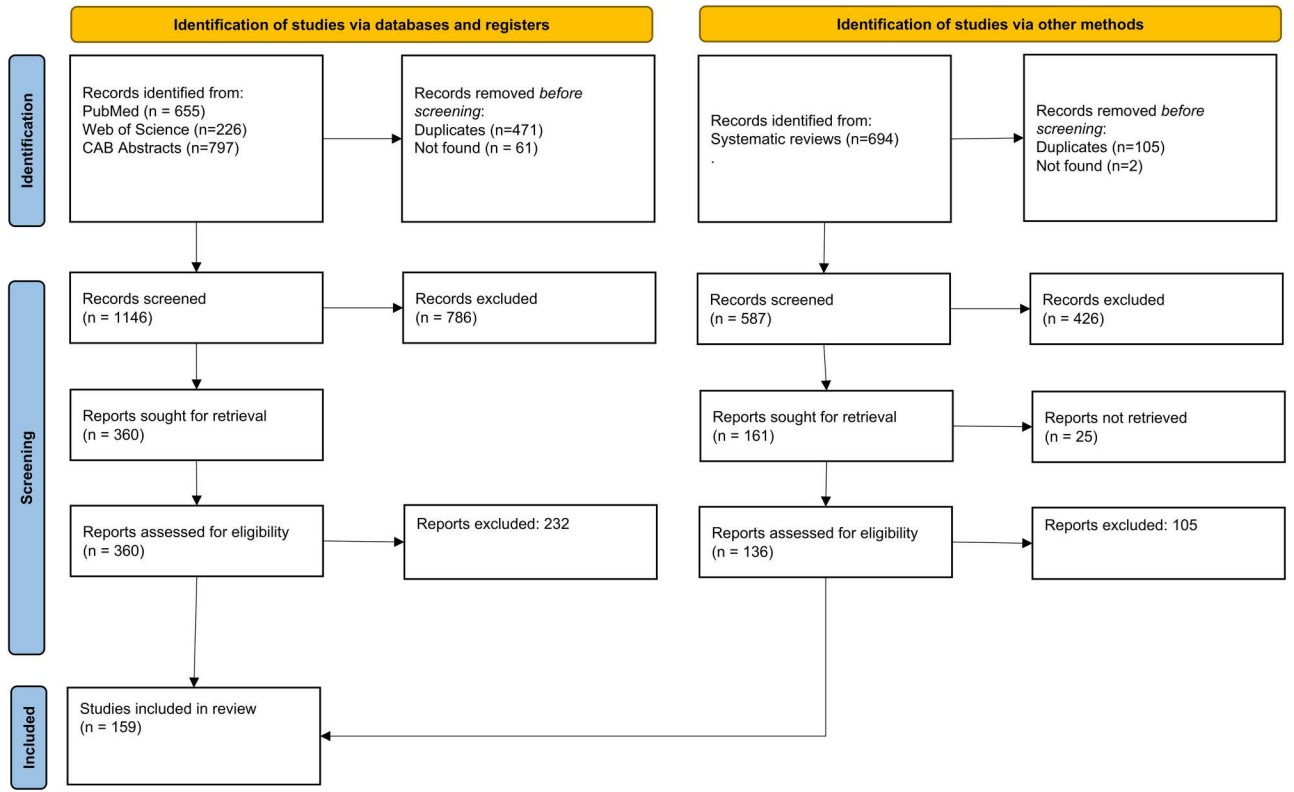

**Fig 2. Flowchart detailing the systematic review of articles on risk factors for TSCT.**

Included studies represented all WHO regions where *T. solium* is endemic (Fig 3A). The African Region had the most included studies (71/159, 45%), with the Region of the Americas a close second (56/159, 35%) (Fig 3A). Included studies were published between 1988 and 2020. Nearly all included studies had a cross-sectional design (143/159, 89.9%).

The 159 included studies contained 876 associations about a proposed cause-effect relationship relevant to *T. solium* epidemiology. Included associations are viewable in S3 File. The quality of evidence for extracted association varied, with weak evidence being the most common (386/876, 44%), followed by moderate (324/876, 37.0%) and strong (166/876, 18.9%) (Fig 3B). Extracted information for the included associations and for the related study, as well as our quality of evidence assessment, are shown in S3 File.

### Distribution of evidence by infection type

Human taeniasis was understudied compared to human and porcine cysticercosis. Out of our 876 extracted associations, 41.6% (365/876) reported on human cysticercosis as outcome, 37.1% (325/876) on porcine cysticercosis as outcome, and only 10.0% (88/876) reported on human taeniasis as outcome. The remaining 12% (105/876) of associations had an outcome that was not an infection state (for example, studying the effect of socioeconomic status on whether latrines were built).

The most studied association was between human gender and human cysticercosis (62/876 associations [7.1%]), closely followed by the association between human age and human cysticercosis (60/876 associations [6.8%]). Age and gender were tied for the most studied risk factors associated with taeniasis (17/88 [19.3%] associations each). Similarly, pig sex was the most-studied risk factor associated with porcine cysticercosis (50/325 [15.4%] associations), with pig age

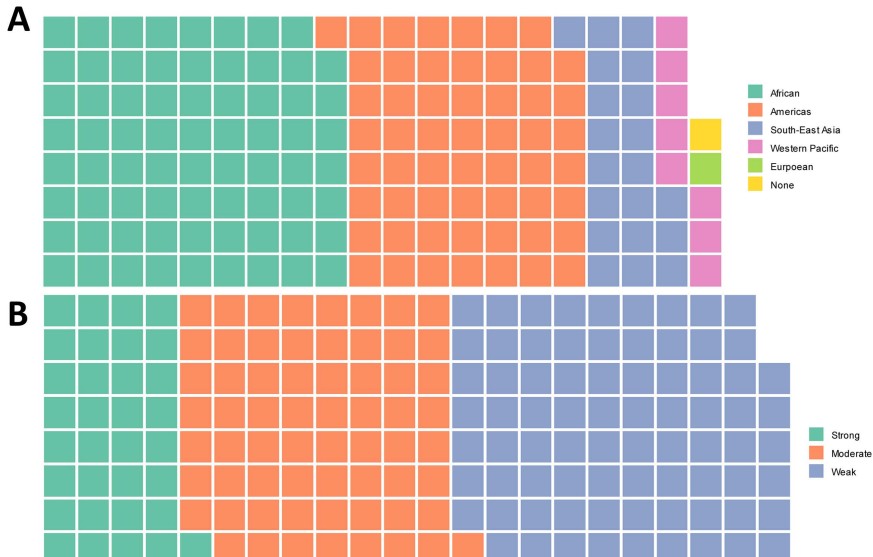

**Fig 3. Waffle charts summarizing the included studies and quality of evidence for the extracted associations, collected in our systematic review. (A)** shows the WHO region [14] where the study was conducted, with each square representing one study. The study marked "none" is a controlled laboratory study that does not have results from a specific region. **(B)** shows the quality of evidence for the extracted associations. Each square represents five extracted associations.

a close second (45/325 [13.9%] associations). None of the associations without a TSCT outcome were well-represented in our study, with the most-studied association (between season and pig roaming) having only six associations extracted (6/876 [0.7%]).

A summary of the frequency of reporting of the associations between different risk factors according to the outcome of interest, stratified by their quality of evidence, is illustrated in Fig 4. Among risk factors with at least five extracted associations, human cysticercosis outcome had the most studied risk factors (18/ 54 [33%]), followed by porcine cysticercosis (16/54 [29.6%]), then taeniasis [11/54 [20.4%], with any other outcome having the fewest risk factors [9/54 [16.7%]) While the quality of evidence was stronger for associations between risk factors, most risk factor-outcome associations were examined by only one or two studies, making interpretation difficult.

Fig 5A–5D shows risk factor-outcome associations with at least five associations extracted. There were no clear trends in the direction of associations when dividing the evidence between those of strong/moderate strength and those of weak strength (Fig 5). A more detailed description of the most frequently reported associations is given below.

### Risk factors common to multiple infection forms

Variables that were examined as potential risk factors for multiple forms of TSCT are summarized below. We present data in alphabetical order of risk factor. All risk factor-outcome associations with at least five measures of the cause-effect association are summarized.

**Age.** Age is not a notable risk factor for human taeniasis, though the evidence is mixed [15–30] (Fig 5B). Most studies found no significant association between human age and cysticercosis, while a little more than one third reported a higher prevalence in older humans [15,16,18,23,25,28–77] (Fig 5A). This tendency was more marked in pigs with a little more than half of the associations indicating a higher prevalence of cysticercosis in older pigs [29,30,46,71,78–105] (Fig 5C). The quality of evidence is usually strong or moderate (Fig 5A–5C) as age is a relatively easy exposure to measure.

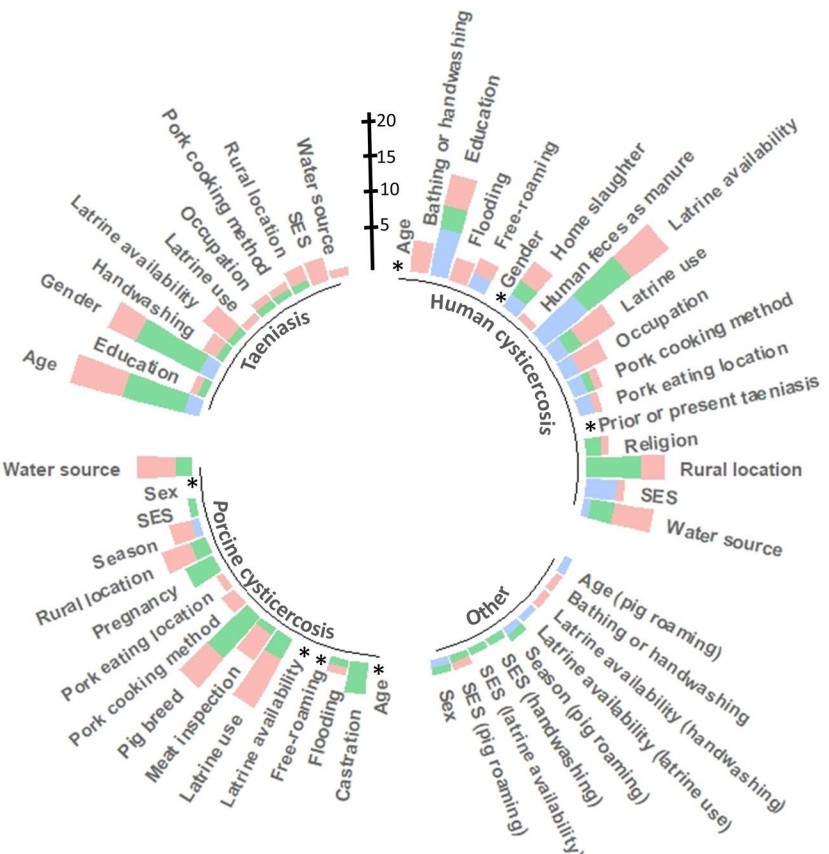

**Fig 4. Circle plot showing the number of associations found for each risk factor.** Associations are sorted by the outcome of the association (human cysticercosis, taeniasis, porcine cysticercosis, and all other outcomes) direction of association found, and the quality of evidence we determined for that association. The blue bar shows the number of associations with strong quality of evidence, the green bar shows the number of associations with moderate quality of evidence, and red shows the number of associations with weak quality of evidence. *Indicates bars set to zero because the large number of reported associations made it impossible to visualize the other associations. Further details on category breakdowns, including those categories set to zero, can be seen in Fig 5A–5D.

In human cysticercosis, the trend towards higher infection with older age is seen the strongest with studies that used an antigen measure of seropositivity (6/11 studies; 55%) [25,32,38,42,46,50,63,64,66,70,74]. In porcine cysticercosis, the trend is higher in antibody tests (6/11; 55%) [29,30,80,83,85,92,94,95,97,102].

**Free-roaming pigs.** Sixteen out of 36 associations found a significant relationship between letting pigs roam free and porcine cysticercosis [30,71,79,84,87,89–91,95,97–102,104,106–120], although the majority of evidence came from associations rated as weak quality (Fig 5C). The association between free-roaming of pigs and human cysticercosis had less evidence, with only one study [56] finding weak-quality evidence of free-roaming of pigs to be a risk factor (Fig 5A). The quality of evidence with this risk factor tends to be weak (Fig 5A and 5C).

**Gender and/or sex.** While most associations evaluated did not identify a significant difference between men and women in the prevalence of human cysticercosis, 13 out of 62 associations suggested a higher prevalence among men [16–18,25,29–34,36,38,40,42–51,54–60,62–76,92,121–129] (Fig 5A). No such differences by gender or sex were observed in studies evaluating taeniasis [15–18,20,24–27,30,43,122,126,130,131] (Fig 5B) or cysticercosis in pigs

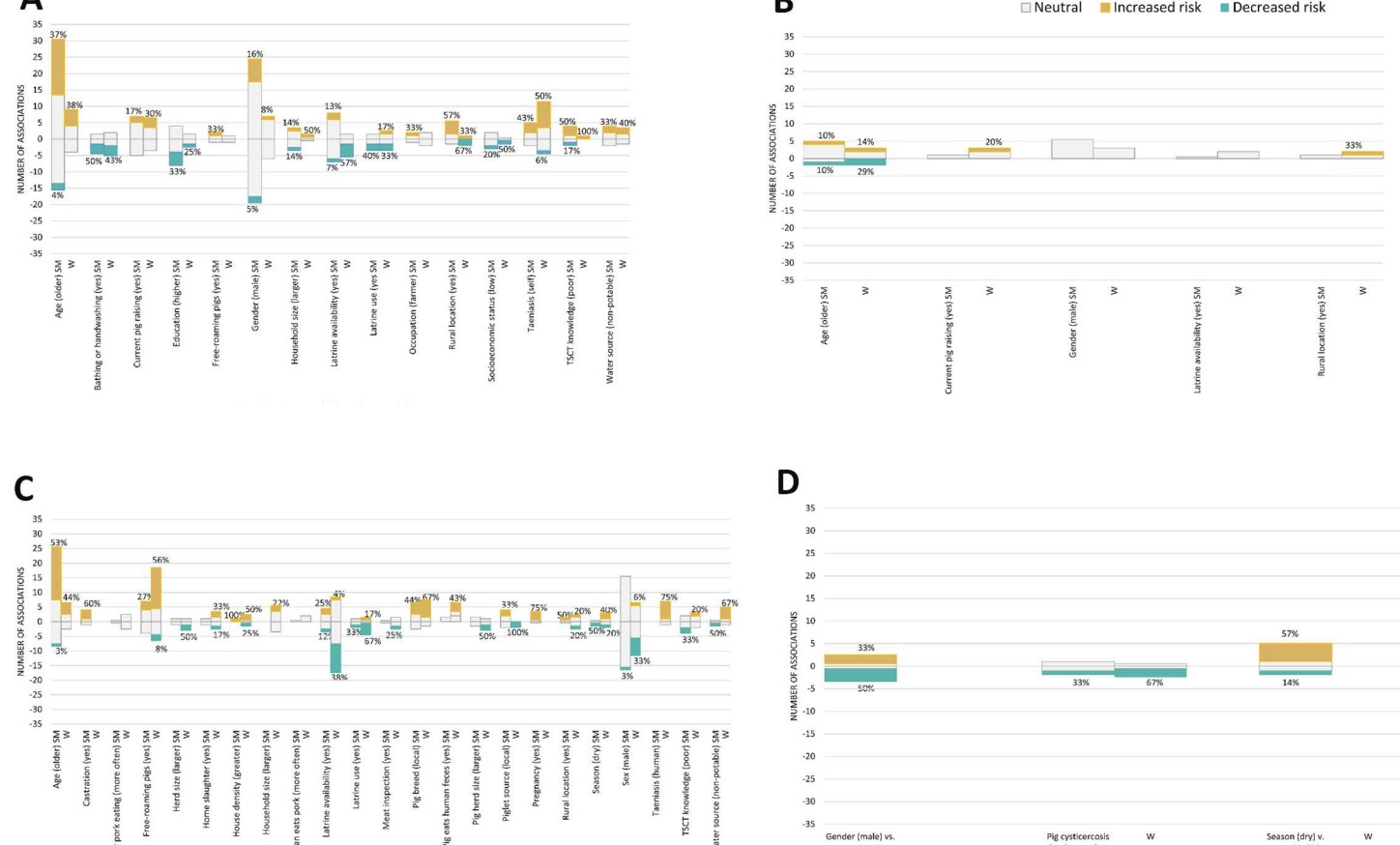

**Fig 5. Bar plots showing the number of associations for each risk factor-outcome association with at least five associations extracted: (A) for those with a human cysticercosis outcome, (B) human taeniasis, and (C) porcine cysticercosis.** Associations are stratified by the direction of the association (increased risk in orange, decreased risk in blue, or neutral in gray). Each chart separates associations rated as strong or moderate (SM) next to associations rated as weak (W). Percentages above each bar represent the percentage of associations in that category that increased the risk (above the 0-axis) or reduced the risk (below the 0-axis) For the sake of visual clarity, the percentage of evidence of neutral quality in each bar is not displayed.

[29,30,46,62,80–89,92,94,95,98,100–105,111,114,119,124,132–145] (Fig 5C). The quality of evidence for this association tends to be strong or moderate (Fig 5A–5C).

**Latrine use and availability.** We are considering both latrine use and latrine availability together as, while they are different concepts, they are highly related. Both latrine use and latrine availability are shown on Fig 5A and 5C. Latrine availability is also shown on Fig 5B. There were not enough extracted associations to visually summarize latrine use as a risk factor for taeniasis.

The majority of the 11 included associations showed no significant association between latrine use on human cysticercosis while three demonstrated a protective effect [31,36,42,46,54,57,63,70,75]. The evidence for an association was stronger for porcine cysticercosis with half of 10 extracted associations supporting latrine use as protective.

Latrine availability is more frequently reported than latrine use, although some studies report on both. Similar results were found between latrine availability and human cysticercosis with five out of 22 extracted associations reporting a protective effect and the remainder showing no significant association

[16,17,31,39,40,42,43,46–48,53,54,56,70,92,128]. The evidence is stronger for pig cysticercosis with nine out of 27 extracted associations observing a significant positive relationship between the lack of latrines and pig cysticercosis [17,39,46,69,79,82–84,89,91,95,98–100,108,109,111–114,116,117].

The quality of evidence for both latrine use and latrine availability tended to be weak (Fig 5A and 5C) The quality of evidence for latrine use is particularly poor (Fig 5A).

**Socioeconomic status.** Socioeconomic status, measured by calculated wealth quintiles [42,70], reported monthly income [58], or community marginalization [34] was measured in relatively few studies. However, using education level as a proxy for socioeconomic status, five out of 16 studies found a protective effect of having a higher level of education on human cysticercosis, although not every association was statistically significant [16,32,36,42,44,46,48,56–58,62,70,71,106,121,146]. Strength of evidence was distributed equally between studies that found a protective effect and studies that did not. Four of the five studies that found a protective effect were of moderate to strong quality while eight of out ten studies that did not find a protective effect were of strong to moderate quality.

**Water source.** Four out of 11 extracted associations suggested that drinking water from an unsafe source increased the frequency of human cysticercosis [16,36,43,46,48,54,57,63,75,121,128]. Likewise, three out of seven associations found that pigs drinking unsafe water were more likely to have cysticercosis [82,84,95,97,99,113,114]. While strong/moderate and weak-quality studies were equally likely to find an association between unsafe water and human cysticercosis (Fig 5A), weak-quality studies were much more likely than studies of strong/moderate quality to find an association between unsafe water and porcine cysticercosis (Fig 5C)

### Risk factors only evaluated *for* human cysticercosis

**Bathing or general hygiene.** Eleven associations from seven studies [15,56,57,68,75,121,125] were extracted. Four of the seven extracted associations identified bathing or general hygiene as protective for human cysticercosis [56,68,121,125], while the remaining three associations were not statistically significant (Fig 5A). Results are similar between studies of strong/moderate quality and studies of weak quality (Fig 5A).

**Taeniasis in the same person.** Twenty-four associations from sixteen studies were extracted. Seven studies found a positive association [15,25,40,43,48,71,122,125] while five found no significant association [18,30,41,44,106]. One study found no cases of cysticercosis among the taeniasis carriers, but there was insufficient data to test for the significance of this association [29,55]. Three studies found a positive association between human cysticercosis and taeniasis in the same person as measured by one metric and no association when measured by the other metric [29,53,68] (Fig 5A). Results are similar between studies of strong/moderate quality and studies of weak quality (Fig 5A).

### Risk factors only evaluated for porcine cysticercosis

**Castration.** Five associations from five studies were extracted. Three of the studies found a higher prevalence of porcine cysticercosis among castrated pigs [84,86,114], while two studies found no significant association [99,141]. All three of the studies that found a significant association share the same lead author, as do both of the studies that found no significant association. All studies evaluating castration as a possible risk factor for human cysticercosis were of strong/moderate quality (Fig 5C).

**Farmer pork eating frequency.** Six associations were extracted from all studies. All six found no significant association between pork eating frequency of the survey respondent and porcine cysticercosis [89,95,98,99,112,147]. Most of these studies were of weak quality (Fig 5C).

### Risk factors of risk factors of TSCT

**Human gender as a cause of pig raising.** Half of the six extracted associations found that women tend to raise pigs more than men [89,141,148], while two extracted associations found that pig farmers tended to be men [145,149]. The

remaining extracted association found no significant relationship between gender and being responsible for raising pigs in the household [150] (Fig 5D). All extracted associations were of strong/moderate quality (Fig 5D). All of the associations that identified a positive association between female gender and pig raising come from countries in the WHO Africa region, but so does one of the associations that found men more likely to be pig-raisers in their community. There is insufficient data to break the results down further by specific country.

## Discussion

This global systematic review considers the factors related to *T. solium* transmission in a One Health manner by considering humans, pigs, and the environment simultaneously. In particular, this approach allowed us to compare the evidence for the same risk factors across the different infective states of *T. solium* in humans and pigs, a task impossible to achieve without a One Health approach. Moreover, this approach was essential to describe potential causal mechanisms at play in the transmission of *T. solium.* Another unique characteristic of this review is its global scope, allowing for exploring regional variations in data and identifying causes with consistent evidence across national borders. Our review also identifies understudied risk factors and those with contradictory information in the literature.

The risk factors of taeniasis are markedly understudied compared to that for human and porcine cysticercosis. The prevalence of taeniasis is low compared to that of human and pig cysticercosis [15,60], making it difficult to reach sufficient power to examine risk factors. Moreover, it has been historically difficult to diagnose *T. solium* taeniasis as part of a large-scale survey. Differentiating *T. solium* from *T. saginata*, a tapeworm with cattle as an intermediate host and humans as definitive host, can only be done by identifying uterine branches from an isolated proglottid or by running molecular tests [151]. Without differentiation, it is impossible for an investigator to determine whether the risk factors of interest are associated with *T. solium* or *T. saginata* induced taeniasis. Finally, human taeniasis incurs far lower burdens of disease than human and porcine cysticercosis, making it harder to obtain research funding. Improvement in access to affordable and valid diagnostics would facilitate the identification of risk factors for taeniasis.

Our review combines a global scope with an interest in all hosts affected by TSCT. Because of our broad focus, we can report on risk factors that have only been studied in one region (e.g., castration of pigs in Latin America) and can extrapolate results from hosts (e.g., sex vs. gender). In particular, we are the first systematic review to conclude that the well-established higher prevalence of cysticercosis in male humans [152] is likely due to gender-related behaviors such as latrine use [153] and not an inherent biological predisposition. In contrast, we found that some modifiable risk factors suck as latrine availability and latrine use were similarly a protective factor in human and porcine cysticercosis. These may be potent targets for eradication efforts as these modifiable risk factors affect two different forms of the disease.

While our included studies cover all WHO regions, most were from the Americas and African Regions. This may be due to sustained interest and research funding obtained by some research groups over decades. As one example, the Cysticercosis Working Group in Peru has published several studies on the epidemiology of TSCT in Peru (see [23,29,47,83,142,154–156] as a non-exhaustive list.)

Several of our risk factor-outcome associations likely reflect indirect causes rather than direct ones. The observed associations between water source and human and pig cysticercosis may be a result of latrine use, as areas without piped water lack flushing toilets. However, *Taenia* eggs can survive in water for weeks [157], so water source may have a direct effect on exposure to *T. solium* eggs. Socioeconomic status likely acts through several mechanisms, including water source (used as part of socioeconomic status determination in [70]) and pig roaming via costs to feed penned pigs [145]. Sex/gender is particularly notable because the trends for human and porcine cysticercosis are in opposite directions. This suggests that the consistently higher cysticercosis prevalence among male humans is from culturally determined factors such as latrine use [153] or pork-eating [158].

Our review considers non-quantitative data, statements of significance and non-significance, and confidence intervals in order to include as much of the evidence as possible. Focus groups in particular, such as used by Thys et al. [148,153],

can provide important information on risk factor-outcome relationships driven by community behavior. Because results from any study are specific to the community being studied, we also considered whether the risk factor-outcome relationships appeared similar in studies in communities from multiple WHO regions when determining whether a purported risk factor was truly a cause of TSCT. Risk factor-outcome relationships that are present in multiple WHO regions are less likely to be present due to culturally-specific confounding factors like social behaviour around latrines (see [153] as an example from one community.)

The quality of available evidence differed greatly across the studies and risk factor-outcome associations included in our review. The quality of evidence for taeniasis was particularly poor due to the low sensitivity and lack of species specificity of many tests such as identification of *Taenia* eggs [159] or self-report of taeniids. More recent tests, such as the rES33 recombinant ELISA, show good sensitivity and specificity without evidence of cross-reactivity [160]. Two studies included in our review used this test [17,18]. If consistently successful, the rES33 test may improve the quality of evidence for these types of studies by more accurately identifying taeniasis cases and reducing false negatives.

The quality of evidence for associations with a porcine cysticercosis outcome also tended to be lower than the quality of evidence for associations with a human cysticercosis outcome. Many of our included studies used lingual exam or meat inspection (see S3 File), and neither is a sensitive measure of infection [161]. Additionally, since pigs are kept in herds, many of the risk factors related to porcine living conditions would affect the entire group and need to be analyzed through a hierarchical model [162]. Our quality assessment graded studies with a combination of two or more serious flaws to be of weak quality [12]. While many risk factors for human cysticercosis also act at the group level (e.g., latrine access, free-roaming of pigs), the use of more accurate diagnostic tests for human cysticercosis kept these studies from being graded as having weak quality of evidence.

Pig age is a difficult variable to assess because the different tests used to identify cysticercosis have different meanings with this risk factor. Antigen tests measure active infection, while antibody tests measure past infections. Animals and humans that are older will have had more time to develop antibodies against *T. solium* and test positive when antibody tests are used. There likely is some true effect of age on cysticercosis in humans given the higher seropositivity among antigen tests, but we cannot tell in pigs given that the length of antigen seropositivity (a few years) is equal to the lifespan of a pig (particularly a pig sent to slaughter for meat.)

This review highlights risk factor-outcome associations which have at least five extracted associations, regardless of the quality of these five extracted associations. For each of our outcomes of interest (human cysticercosis, human taeniasis, porcine cysticercosis, and associations between risk factors) there were several associations with few or even a single extracted association (e.g., the association between parents' education level and routine deworming of children in Openshaw et al [31]). These associations, found in S3 File, need further study.

One limitation of our review is that most identified articles reported on studies applying a cross-sectional design. It can be difficult to infer causality based on such a design because purported causes and their effects are measured at the same time point, making evaluation of temporality challenging [163]. Even with antigen tests that measure active infection, we cannot rule out that modifiable risk factors such as latrine use may have changed between the time of infection and the time of the study (for example, if a program built latrines in the village recently.) This problem of determining temporality is often compounded by the use of antibody-detecting serological tests, which measure prior exposure rather than active infection, thereby further obscuring temporal relationships. Nonetheless, since most individuals with TSCT infections are asymptomatic [85], it remains unlikely that infection itself substantially influenced behavioral patterns in the participants in our included studies.

A second limitation of our review is that we might not have captured all relevant literature in the field. First, the use of the term "epidemiology" in our search phrase may have excluded relevant studies that explored associations among risk factors and the TSCT outcome without mentioning the term "epidemiology" in their publication. However, we feel we have partly compensated for this by screening all articles cited in systematic reviews identified through our database search.

Second, because we restricted inclusion to peer-reviewed journal articles, we may have missed relevant evidence from the grey literature (e.g., reports, theses, conference proceedings), potentially introducing publication bias and limiting the generalisability of our findings.

In summary, we have presented an overview of the current evidence for risk factors of *T. solium* cysticercosis and taeniasis in pigs and humans. Ours is one of the only reviews to consider risk factors of all three disease forms in the same study, while also considering results from studies from around the world. This approach permits a comprehensive assessment of risk factors for different states of a same infection in different hosts, which in turn can help determine causal mechanisms at play.

## Supporting information

**S1 File. Modifications: Modifications made to the EPHPP tool.**
(DOCX)

**S2 File. Grading Diagnostics: Grading criteria for validity and reliability of questionnaires and diagnostic tests.**
(XLSX)

**S3 File. Extracted Data Sheet: Extracted data sorted by exposure and outcome variables.**
(XLSX)

**S1 Checklist. PRISMA abstract checklist.**
(DOCX)

**S2 Checklist. PRISMA checklist for this systematic review.**
(DOCX)

## Author contributions

**Conceptualization:** Elias Jackson, Amanda Janitz, Hélène Carabin.

**Data curation:** Elias Jackson, Veronique Dermauw, Guy Eyaba Tchamdja, Mohammad Shah Jalal, Katrina Di Bacco.

**Formal analysis:** Elias Jackson, Veronique Dermauw, Pierre Dorny, Amanda Janitz, Hélène Carabin.

**Funding acquisition:** Elias Jackson, Hélène Carabin.

**Investigation:** Veronique Dermauw.

**Methodology:** Elias Jackson, Veronique Dermauw, Guy Eyaba Tchamdja, Amanda Janitz, Hélène Carabin.

**Supervision:** Amanda Janitz, Hélène Carabin.

**Writing – original draft:** Elias Jackson.

**Writing – review & editing:** Elias Jackson, Veronique Dermauw, Mohammad Shah Jalal, Amanda Janitz, Hélène Carabin.

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
