## [Decision Letter · Decision Letter 0]

11 Sep 2025

PNTD-D-25-00379

Elucidating the causal mechanisms of Taenia solium transmission in humans, pigs and the environment: a global systematic review.

Dear Dr. Carabin,

Thank you for submitting your manuscript to PLOS Neglected Tropical Diseases. After careful consideration, we feel that it has merit but does not fully meet PLOS Neglected Tropical Diseases's publication criteria as it currently stands. Therefore, we invite you to submit a revised version of the manuscript that addresses the points raised during the review process.

Please submit your revised manuscript within 60 days Nov 10 2025 11:59PM. If you will need more time than this to complete your revisions, please reply to this message or contact the journal office at plosntds@plos.org. Please include the following items when submitting your revised manuscript:

We look forward to receiving your revised manuscript.

Kind regards,

Benn Sartorius, PhD

Section Editor

Shaden Kamhawi

co-Editor-in-Chief

Paul Brindley

co-Editor-in-Chief

**Journal Requirements:**

1) Please upload all main figures as separate Figure files in .tif or .eps format. For more information about how to convert and format your figure files please see our guidelines:

2) We have noticed that you have uploaded Supporting Information files, but you have not included a list of legends. Please add a full list of legends for your Supporting Information files after the references list.

3) Some material included in your submission may be copyrighted. According to PLOSu2019s copyright policy, authors who use figures or other material (e.g., graphics, clipart, maps) from another author or copyright holder must demonstrate or obtain permission to publish this material under the Creative Commons Attribution 4.0 International (CC BY 4.0) License used by PLOS journals. Please closely review the details of PLOSu2019s copyright requirements here: PLOS Licenses and Copyright. If you need to request permissions from a copyright holder, you may use PLOS's Copyright Content Permission form.

Potential Copyright Issues:

i) Figure 1. Please confirm whether you drew the images / clip-art within the figure panels by hand. If you did not draw the images, please provide (a) a link to the source of the images or icons and their license / terms of use; or (b) written permission from the copyright holder to publish the images or icons under our CC BY 4.0 license. Alternatively, you may replace the images with open source alternatives. See these open source resources you may use to replace images / clip-art:

4) Please amend your detailed Financial Disclosure statement. This is published with the article. It must therefore be completed in full sentences and contain the exact wording you wish to be published.

2) If any authors received a salary from any of your funders, please state which authors and which funders.

5) Please ensure that the PRISMA flowchart describing the included/excluded literature is uploaded as Figure 1. Information about the PRISMA guidance, and blank flowcharts and checklists, can be found here: http://www.prisma-statement.org/.

6) As required by our policy on Data Availability, please ensure your manuscript or supplementary information includes the following:

This information can be included in the main text, supplementary information, or relevant data repository.

**Comments to the Authors:**

**Please note that one review is uploaded as an attachment.**

**Reviewers' Comments:**

Reviewer's Responses to Questions

**Key Review Criteria Required for Acceptance?**

**Methods**

-Are the objectives of the study clearly articulated with a clear testable hypothesis stated?

-Is the study design appropriate to address the stated objectives?

-Is the population clearly described and appropriate for the hypothesis being tested?

-Is the sample size sufficient to ensure adequate power to address the hypothesis being tested?

-Were correct statistical analysis used to support conclusions?

-Are there concerns about ethical or regulatory requirements being met?

Reviewer #1: see comments on the attached reviewer comments document

Reviewer #2: -Are the objectives of the study clearly articulated with a clear testable hypothesis stated?

The present study is a systematic review, and the objective was to identify the risk factors of T. solium infection in humans, pigs, and the environment. The authors set a goal and sought to determine whether the risk factors for acquiring human cysticercosis, porcine cysticercosis, or Taenia solium taeniasis have been adequately studied. They also sought to determine whether any of the risk factors studied for one form of the disease could be applied to another. Whether there was a connection between the objective and the hypothesis for this type of review.

Is the study design appropriate to address the stated objectives?

Indeed, the study design was appropriate for meeting the objectives of this systematic review. The authors conducted multiple literature searches and followed the PRISMA guidelines for data analysis and the presentation of results.

Is the population clearly described and appropriate for the hypothesis being tested?

Yes, the population was clearly described. The authors mention that they conducted a bibliographic review of articles on the epidemiology of Taenia solium up to May 2020. They consulted the PubMed, Web of Science, and CABAbstracts data bases without filters and in different languages. In the results section, they report that they found 2,113 studies that met the search terms, that 470 met the criteria for full reviews, and that only 158 met all the inclusion criteria.

Is the sample size sufficient to ensure adequate power to address the hypothesis being tested?

The sample size is sufficient. Systematic reviews have been conducted with fewer than 50 articles reviewed, and they analyzed 158 articles

Were correct statistical analysis used to support conclusions?

This is a systematic review that does not require rigorous statistical analysis, so this section is appropriate. The information found in the 158 publications was not reported homogeneously, so the authors extracted the necessary information and found 833 associations, some stronger and others weaker, results that they qualitatively describe.

-Are there concerns about ethical or regulatory requirements being met? -

There are no concerns about compliance with ethical or regulatory requirements. The procedures followed are clearly explained. This is a systematic review

**Results**

-Does the analysis presented match the analysis plan?

-Are the results clearly and completely presented?

-Are the figures (Tables, Images) of sufficient quality for clarity?

Reviewer #1: see attached document

Reviewer #2: Does the analysis presented match the analysis plan?

The analysis presented is consistent with the objective of the study

-Are the results clearly and completely presented?

Yes, the results are clearly presented

Are the figures (Tables, Images) of sufficient quality for clarity? Figures are not of enough quality for clarity

Figure 1: The life cycle of Taenia solium.

The life cycle of Taenia solium must be modified. Several freely licensed images can be used without any problem. But if authors are going to make their own drawing of the parasite's life cycle, they can use different platforms like BioRender, Canva, or other systems to make images more similar to the real thing and make it clear what they're pointing out. The cysticerci, eggs, Taenia, and the pig don't look alike at all.

Figure 2: A PRISMA flowchart detailing their literature review.

Figure 2 needs more definition; the writing in the white and blue boxes is barely legible. Although it's a program-generated image, it could be improved, as it's not readable at normal size, let alone when enlarged.

**Conclusions**

-Are the conclusions supported by the data presented?

-Are the limitations of analysis clearly described?

-Do the authors discuss how these data can be helpful to advance our understanding of the topic under study?

-Is public health relevance addressed?

Reviewer #1: (No Response)

Reviewer #2: -Are the conclusions supported by the data presented?

Yes, conclusions are supported by the data presented in the revision

-Are the limitations of analysis clearly described?

The authors describe a limitation in their study in lines 351-355. The explanation is brief, and apparently, they were able to solve it adequately, on line 354

-Do the authors discuss how these data can be helpful to advance our understanding of the topic under study?

Yes, the authors discuss the results of the literature review. They identified new risk factors and confirmed those previously defined for several decades. Throughout the text, they discuss the various risk factors associated with contracting taeniasis, human cysticercosis, and porcine cysticercosis.

-Is public health relevance addressed?

Yes, reference was made to taeniasis, human cysticercosis, and porcine cysticercosis. The problems each disease causes in the host and its impact on the global population. The authors also investigated and defined risk factors for developing the disease in both hosts.

**Editorial and Data Presentation Modifications?**

Reviewer #1: Better take way messages should be crafted.

Reviewer #2: Authors should note the publication by Ngwili N, et al. (2021, A qualitative assessment of the context and enabling environment for the control of Taenia solium infections in endemic settings. PLoS Negl Trop Dis 15(6): e0009470. https://doi.org/10.1371/journal.pntd.0009470) in presenting their review as “the first to consider all hosts and diseases caused by Taenia solium. See abstract and general comments.

**Summary and General Comments**

Reviewer #1: (No Response)

Reviewer #2: The authors conducted a systematic review of the Taenia solium parasite, using the terms “Taenia solium” AND “epidemiology”) They looked for associations between risk factors for taeniasis, human cysticercosis, and porcine cysticercosis. They found a high number of associations, although most of them were weak. Despite this, they found important findings, as reported in lines 331-334.

However, the authors mention that there are no systematic reviews of TSCT that consider causes of human cysticercosis, human taeniasis, and porcine cysticercosis. Recent reviews summarize the prevalence and geographic distribution of disease in one species, or only consider causation in one species and one region (lines 72-75).

However, this justification is not entirely correct, as there are some systematic reviews that address this parasitic disease, taking into account human and porcine taeniasis and cysticercosis. An example of this is the publication by Ngwili N et al. (2021) A qualitative assessment of the context and enabling environment for the control of Taenia solium infections in endemic settings. PLoS Negl Trop Dis 15(6): e0009470. https://doi.org/10.1371/journal.pntd.0009470

In this systematic review, the authors considered the two hosts and three diseases (taeniasis, human cysticercosis, and porcine cysticercosis), and the literature search was global, from 1950 to 2019. Although they do not publicize their work with a One Health approach, Ngwili et al. took into account the same considerations as the present review.

I believe that this previous work should be taken into account since the present manuscript (Elias Jackson et al.) is not the first systematic review to focus on the three diseases and two hosts involved in the life cycle of Taenia solium.

PLOS authors have the option to publish the peer review history of their article (what does this mean?). If published, this will include your full peer review and any attached files.

Reviewer #1: No

Reviewer #2: No

**Figure resubmission:**
---

## [Decision Letter · Decision Letter 1]

20 Apr 2026

Dear Dr. Jackson,

We are pleased to inform you that your manuscript 'Elucidating the causal mechanisms of Taenia solium transmission in humans, pigs and the environment: a global systematic review.' has been provisionally accepted for publication in PLOS Neglected Tropical Diseases.

Best regards,

Joseph Raymond Zunt, PhD

Academic Editor

Max Thomas Eyre, PhD

Section Editor

Benn Sartorius

%CORR_ED_EDITOR_ROLE%

Shaden Kamhawi

co-Editor-in-Chief

Paul Brindley

co-Editor-in-Chief

Reviewer’s Responses to Questions

**Key Review Criteria Required for Acceptance?**

**Methods**

-Are the objectives of the study clearly articulated with a clear testable hypothesis stated?

-Is the study design appropriate to address the stated objectives?

-Is the population clearly described and appropriate for the hypothesis being tested?

-Is the sample size sufficient to ensure adequate power to address the hypothesis being tested?

-Were correct statistical analysis used to support conclusions?

-Are there concerns about ethical or regulatory requirements being met?

Reviewer #2: Since the first version of the manuscript, authors set a goal and sought to determine whether the risk factors for acquiring human cysticercosis, porcine cysticercosis, or Taenia solium taeniasis have been adequately studied. They also sought to determine whether any of the risk factors studied for one form of the disease could be applied to another. Whether there was a connection between the objective and the hypothesis for this type of review.

Indeed, the study design was appropriate for meeting the objectives of this systematic review.

Yes, the population was clearly described. The authors mention that they conducted a bibliographic review of articles on the epidemiology of Taenia solium up to May 2020. They consulted the PubMed, Web of Science, and CABAbstracts data bases without filters and in different languages. In the results section, they report that they found 2,113 studies that met the search terms, that 470 met the criteria for full reviews, and that only 158 met all the inclusion criteria.

The sample size is sufficient. Systematic reviews have been conducted with fewer than 50 articles reviewed, and they analyzed 159 articles

This is a systematic review that does not require rigorous statistical analysis, so this section is appropriate. The information found in the 159 publications was not reported homogeneously, so the authors extracted the necessary information and found 833 associations, some stronger and others weaker, results that they qualitatively describe.

There are no concerns about compliance with ethical or regulatory requirements. The procedures followed are clearly explained. This is a systematic review

**Results**

-Does the analysis presented match the analysis plan?

-Are the results clearly and completely presented?

-Are the figures (Tables, Images) of sufficient quality for clarity?

Reviewer #2: Does the analysis presented match the analysis plan?

The analysis presented is consistent with the objective of the study

-Are the results clearly and completely presented?

Yes, the results are clearly presented

Are the figures (Tables, Images) of sufficient quality for clarity?

In this second version, the authors made notable changes to the figures, especially Figures 1 and 2. The life cycle of Taenia solium is now clearer and more understandable. Figure 2 is more legible.

**Conclusions**

-Are the conclusions supported by the data presented?

-Are the limitations of analysis clearly described?

-Do the authors discuss how these data can be helpful to advance our understanding of the topic under study?

-Is public health relevance addressed?

Reviewer #2: Are the conclusions supported by the data presented?

Yes, conclusions are supported by the data presented in the revision

-Are the limitations of analysis clearly described?

In this second version, the authors described in more detail the limitations they encountered in their bibliographic research.

Do the authors discuss how these data can be helpful to advance our understanding of the topic under study?

Yes, the authors discuss the results of the literature review. They identified new risk factors and confirmed those previously defined for several decades. Throughout the text, they discuss the various risk factors associated with contracting taeniasis, human cysticercosis, and porcine cysticercosis.

-Is public health relevance addressed?

Yes, reference was made to taeniasis, human cysticercosis, and porcine cysticercosis. The problems each disease causes in the host and its impact on the global population. The authors also investigated and defined risk factors for developing the disease in both hosts.

**Editorial and Data Presentation Modifications?**

Reviewer #2: The authors took into account observations on other publications related to the topic, that were mentioned in the previous review

**Summary and General Comments**

Reviewer #2: This second version of the manuscript reads much better, and the authors took into account the observations of the different reviewers to enrich the manuscript, discuss it more thoroughly, and emphasize the most outstanding results.

PLOS authors have the option to publish the peer review history of their article (what does this mean?). If published, this will include your full peer review and any attached files.

Reviewer #2: No